# From Stoner to local moment magnetism in atomically thin Cr$_2$Te$_3$

Yong Zhong [1,2,3,10] ✉, Cheng Peng[2,10], Haili Huang[4,10], Dandan Guan [2,3,4] ✉, Jinwoong Hwang[1,2,5], Kuan H. Hsu[6], Yi Hu [6], Chunjing Jia[2,7], Brian Moritz[2], Donghui Lu [8], Jun-Sik Lee [8], Jin-Feng Jia [4], Thomas P. Devereaux [2,6], Sung-Kwan Mo [1] ✉ & Zhi-Xun Shen [2,3,9] ✉

The field of two-dimensional (2D) ferromagnetism has been proliferating over the past few years, with ongoing interests in basic science and potential applications in spintronic technology. However, a high-resolution spectroscopic study of the 2D ferromagnet is still lacking due to the small size and air sensitivity of the exfoliated nanoflakes. Here, we report a thickness-dependent ferromagnetism in epitaxially grown Cr$_2$Te$_3$ thin films and investigate the evolution of the underlying electronic structure by synergistic angle-resolved photoemission spectroscopy, scanning tunneling microscopy, x-ray absorption spectroscopy, and first-principle calculations. A conspicuous ferromagnetic transition from Stoner to Heisenberg-type is directly observed in the atomically thin limit, indicating that dimensionality is a powerful tuning knob to manipulate the novel properties of 2D magnetism. Monolayer Cr$_2$Te$_3$ retains robust ferromagnetism, but with a suppressed Curie temperature, due to the drastic drop in the density of states near the Fermi level. Our results establish atomically thin Cr$_2$Te$_3$ as an excellent platform to explore the dual nature of localized and itinerant ferromagnetism in 2D magnets.

Although the quantum theory of magnetism was established in the early days of quantum mechanics, the localized vs. delocalized conundrum remains controversial in many magnetic materials[1]. Based on the itinerant picture, the Stoner model successfully predicts the exchange splitting of electron bands and non-integer magnetic moments in ferromagnetic metals[2]. On the contrary, the Heisenberg model is usually used to interpret the antiferromagnetic and ferromagnetic order of local moments in magnetic insulators[3]. In the 1960s, the Hubbard model was proposed to combine the "localized" and "delocalized" pictures under a unified framework[4], which principally resolves the dual nature

of electrons in magnets. However, an exact solution to the Hubbard model is impossible[5]. Instead, as a simplification of the Hubbard model, the spin-fluctuation method is widely used to qualitatively describe the magnetic properties in half-metals[6]. Among the rich branches of magnetism, exploring 2D magnetic order is extremely important to the development of phase transition theory and statistical physics. Although Mermin-Wagner theorem prohibits the long-range 2D magnetic order at any finite temperature in an isotropic spin orientation case[7], Onsager and Kosterlitz-Thouless prove the existence of 2D magnetism in a lower symmetry (spin anisotropy) system[8,9].

[1]Advanced Light Source, Lawrence Berkeley National Laboratory, Berkeley, CA 94720, USA. [2]Stanford Institute for Materials and Energy Sciences, SLAC National Accelerator Laboratory, Menlo Park, CA 94025, USA. [3]Department of Applied Physics, Stanford University, Stanford, CA 94305, USA. [4]Key Laboratory of Artificial Structures and Quantum Control (Ministry of Education), TD Lee Institute, School of Physics and Astronomy, Shanghai Jiao Tong University, 200240 Shanghai, China. [5]Department of Physics and Institute of Quantum Convergence Technology, Kangwon National University, Chucheon 24341, Republic of Korea. [6]Department of Materials Science and Engineering, Stanford University, Stanford, CA 94305, USA. [7]Department of Physics, University of Florida, Gainesville, FL 32611, USA. [8]Stanford Synchrotron Radiation Lightsource, SLAC National Accelerator Laboratory, Menlo Park, CA 94025, USA. [9]Department of Physics, Stanford University, Stanford, CA 94305, USA. [10]These authors contributed equally: Yong Zhong, Cheng Peng, Haili Huang. ✉e-mail: ylzhong@stanford.edu; ddguan@sjtu.edu.cn; skmo@lbl.gov; zxshen@stanford.edu

The research on 2D ferromagnetism has advanced rapidly thanks to the recent development of fabrication methods and characterization tools[10–12]. Among the material library, the most studied are $Cr_2Ge_2Te_6$, $CrI_3$, and $Fe_3GeTe_2$[13–18]. Although the 2D ferromagnetism in insulating $Cr_2Ge_2Te_6$ and $CrI_3$ is well represented by Heisenberg or Ising-like models[13,14,17,18], no consensus has been achieved for metallic $Fe_3GeTe_2$. While transport measurements claim the itinerant ferromagnetism[15,16], an angle-resolved photoemission spectroscopy (ARPES) study shows the absence of exchange splitting energy across the Curie temperature ($T_C$)[19]. Additionally, scanning tunneling spectroscopy measurement finds a Kondo lattice-like behavior in the ferromagnetic state[20], implying a dual nature of localized and itinerant electrons. A similar situation also occurs in the prototypical ferromagnet $SrRuO_3$[21,22]. These examples demonstrate that ferromagnetic metals contain more physics beyond the simple Stoner model. Furthermore, despite extensive efforts to explore the corresponding macroscopic properties, there is still no ARPES study to elucidate the underlying band structures of 2D magnets, which hinders the comprehensive understanding of the microscopic mechanism. Therefore, large-area atomically thin ferromagnets are highly desired, not only to unveil the nature of 2D ferromagnetism, but also for future practical applications.

$Cr_2Te_3$ is a ferromagnetic metal with $T_C = 170$ K[23]. The easy axis is along the out-of-plane direction. It consists of Cr-deficient ($Cr_1$ atom) and Cr-filled layers ($Cr_2$ and $Cr_3$ atoms), as shown in Fig. 1a. Each Cr atom is surrounded by six Te atoms, forming a corner-shared octahedron. According to the ionic picture, the covalent state of Cr is expected to be +3 with an electronic configuration $3d^3$. In an octahedral structure, the crystal field splits the five $d$ orbitals into a higher energy $e_g$ doublet and a lower energy $t_{2g}$ triplet. Thus, the $Cr^{3+}$ ion has a total spin of $S = 3/2$, with three electrons in the $t_{2g}$ manifold to satisfy Hund's rule. Experimentally, the saturated magnetic moment is

2.65 $\mu_B$/Cr[24]. This non-integer and large magnetization supports a picture of itinerant ferromagnetism in $Cr_2Te_3$. In addition, neutron scattering measurements show $-0.14$ $\mu_B$, 2.78 $\mu_B$ and 2.53 $\mu_B$ for $Cr_1$, $Cr_2$, and $Cr_3$, indicating the essential role of the Cr-filled layer in driving the long-range ferromagnetic order in $Cr_2Te_3$. $Cr_2Te_3$ thin films have been fabricated by molecular beam epitaxy (MBE) method on silicon and sapphire substrates[25,26], displaying a comparable $T_C$ to the single crystals as the film thickness exceeds 4 nm. However, to our best knowledge, there is no exploration of the ferromagnetism in the atomically thin 2D limit.

Here, we report a dimensionality-induced ferromagnetic ground state transition in the atomically thin $Cr_2Te_3$. We used MBE to prepare $Cr_2Te_3$ thin films. 2D ferromagnetism was observed in the samples with thickness from monolayer (ML) to 6 ML, although the $T_C$ drops to nearly half of the bulk value in the monolayer case. To understand this drastic change, ARPES, x-ray absorption spectroscopy (XAS), and density functional theory (DFT) calculations were utilized to simultaneously track the thickness-dependent electronic structures in the $Cr_2Te_3$ system. A suppression of density of states (DOS) near the Fermi level ($E_F$) is consistent with the lower $T_C$ in 1 ML $Cr_2Te_3$. Most importantly, in contrast to the Stoner-type ferromagnetism exhibited in thicker films, there is a fundamental ground state transition to the localized magnetism in the 2D limit. Hence, our study provides another important example of novel physics at the atomically thin limit, as the indirect to direct gap transition in $MoSe_2$[27], and the Weyl semimetal to quantum spin Hall insulator transition in $WTe_2$[28].

## Results
### From 3D to 2D ferromagnetism
Figure 1b displays the reflection high-energy electron diffraction (RHEED) patterns during the process of sample preparation, in which the sharp streaks indicate the epitaxial nature of the thin film. The

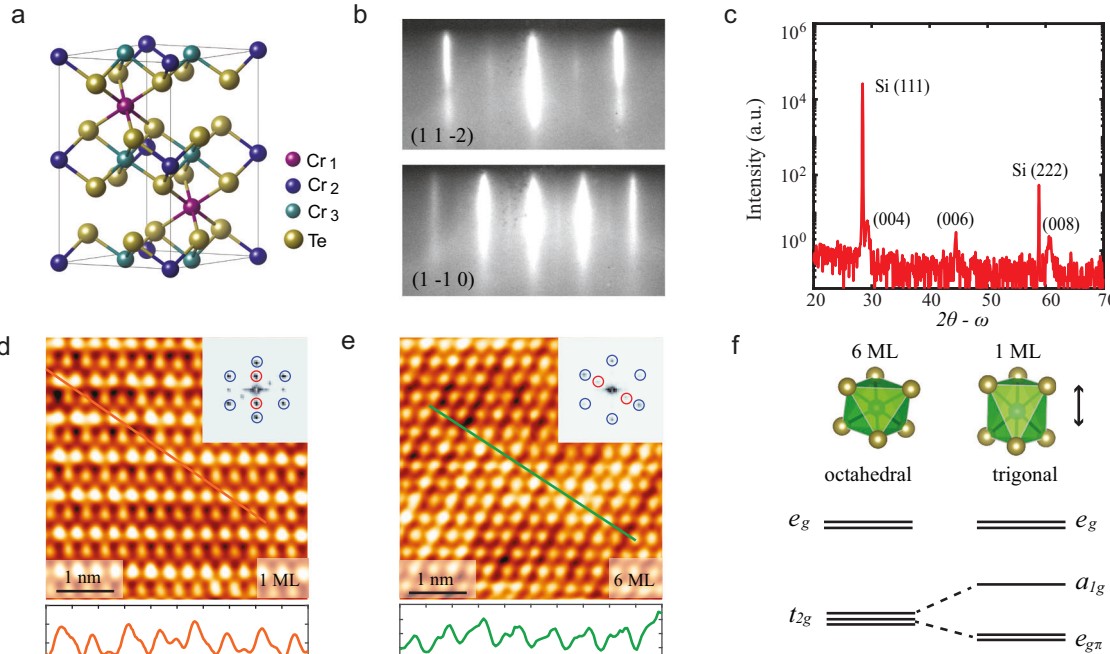

**Fig. 1 | Preparation and structural characterization of $Cr_2Te_3$ thin films on Si (111) substrate. a** Crystal structure of $Cr_2Te_3$. Cr-filled layer (consisting of $Cr_2$ and $Cr_3$ atoms) and Cr-deficient layer (consisting of $Cr_1$ atoms) stack alternatively. **b** RHEED patterns from $Cr_2Te_3$ along (1 1 −2) and (1 −1 0) orientations of Si (111) substrate, respectively. **c** Wide-angle 2θ-ω x-ray diffraction. Except for the typical peak from the substrate, the $Cr_2Te_3$ (004), (006), and (008) peaks are clearly observed. **d, e** Upper panels: atomic resolution images of 1 ML and 6 ML thin films. Lower panels: the corresponding line profiles along the orange and green cuts,

respectively. Insets show the fast Fourier transformation analysis. Blue circles are the signals of the unit cell, whereas red circles denote the 2 × 2 reconstruction. Scanning parameters are $V = -10.9$ mV, $I = 179$ pA for 1 ML sample, and $V = 39.3$ mV, $I = 100$ pA for 6 ML sample. **f** The local $CrTe_6$ cluster transforms from octahedral symmetry in 6 ML sample to trigonal symmetry in 2D limit. Accordingly, the degenerate $t_{2g}$ bands further split into $a_{1g}$ and $e_{g\pi}$ bands, as shown in the schematic diagram.

single-crystalline $Cr_2Te_3$ phase was confirmed by x-ray diffraction measurement on 6 ML sample. Except for the substrate peaks, a set of $Cr_2Te_3$ peaks are clearly observed in the 2θ-ω scan (Fig. 1c). In-situ scanning tunneling microscopy measurement further illustrates the atomic-scale structure of 1 ML and 6 ML samples. Both line-profile cuts and fast Fourier transformation analysis are used to determine the in-plane lattice parameters: the lattice constant of the 1 ML sample is 6.32 Å (Fig. 1d), which is nearly the same as that of the Si (111) substrate. The lattice parameter however grows to 6.81 Å in the 6 ML sample (Fig. 1e), closer to the value of bulk $Cr_2Te_3$. In addition, the out-of-plane lattice parameter in the 1 ML sample is 13.3 Å, which is 7% larger than that of the 6 ML sample (Supplementary Fig. 1). All the above observations indicate a significant strain effect in the 1 ML sample, which transforms the pristine $CrTe_6$ octahedral structure to the trigonal symmetry, further splitting the degenerate $t_{2g}$ bands into $a_{1g}$ and $e_{g\pi}$ bands (Fig. 1f). Fourier transformation analysis shows clear 1 × 2 surface reconstructions on the 1 ML and 6 ML samples, consistent to the previous study[25]. Combining the in-plane and out-of-plane lattice information, pure $Cr_2Te_3$ structure is unambiguously obtained in our thin films, without any impurities or phase separation. Temperature-dependent magnetization of 6 ML sample was measured by superconducting quantum interference device (SQUID) in Supplementary Fig. 2, showing the same $T_C$ as that in literatures[25,26]. However, the elusive ferromagnetic signal of 1 ML sample requires a technique of better sensitivity than that provided by conventional SQUID.

X-ray magnetic circular dichroism (XMCD) is a unique element-specific technique to explore the local electronic and magnetic structures of $3d$ transition-metal compounds, which is the difference spectrum of two polarized absorption spectra taken under an external magnetic field. This method has become a standard tool for investigating ferromagnetism in 2D magnets[15,16,18], even with small net magnetic moments. Figure 2a, b display the left-/right-circularly polarized absorption spectra ($\rho^+$ and $\rho^-$) and the corresponding difference spectra (XMCD = $\rho^+ - \rho^-$) on the 1 ML and 3 ML $Cr_2Te_3$ samples. For the 3 ML sample, an obvious XMCD signal around Cr $L_{2,3}$ edges (located at 575 eV and 583 eV) is observed corresponding to long-range ferromagnetic order, consistent with the reported moment of 2.65 $\mu_B$. In the monolayer sample case, the ferromagnetic feature is still clear although the dichroic signal is weak. It implies the existence of 2D ferromagnetism in the $Cr_2Te_3$ system.

In addition, temperature-dependent XMCD measurements were performed on the 1 ML, 3 ML, and 6 ML samples, as shown in Fig. 2c–e. We defined $T_C$ as the temperature at which the XMCD signal disappears. While the same $T_C$ (=170 K) was demonstrated in the 3 ML and 6 ML samples, we observed that a drastic suppression of $T_C$ (=90 K) occurs in the monolayer case. Similar phenomena were observed in other 2D magnets, where the 2D ferromagnetic transition temperature drops to half or one-third of the bulk value[13–16]. Furthermore, we can use the ratio of the XMCD and XAS signals to depict the stiffness of ferromagnetism, which is proportional to the amplitude of magnetization[18]. Figure 2f displays the temperature-dependent XMCD percentage on the 1 ML, 3 ML, and 6 ML samples. The signal in the monolayer sample is only 5% of that in the 6 ML sample (nominal value is 1/6), indicating a magnetic ground state change from 3D to 2D limit, which will be discussed later.

## From Stoner to Heisenberg ferromagnetism

Temperature-dependent ARPES measurements were performed to investigate the ferromagnetic structure in the 2D limit. For simplicity, we choose 6 ML and 1 ML samples as representatives to illustrate the crossover from 3D to 2D (Fermi surface and band structure of the 3 ML sample are shown in Supplementary Fig. 3). Figure 3a, b display the Fermi surface maps of the 6 ML and 1 ML $Cr_2Te_3$ samples. Circular hole pockets are clearly observed around the Brillouin zone center $\Gamma$ point. By varying the temperature below and above $T_C$, the electronic structures of the ferromagnetic and paramagnetic states are systematically

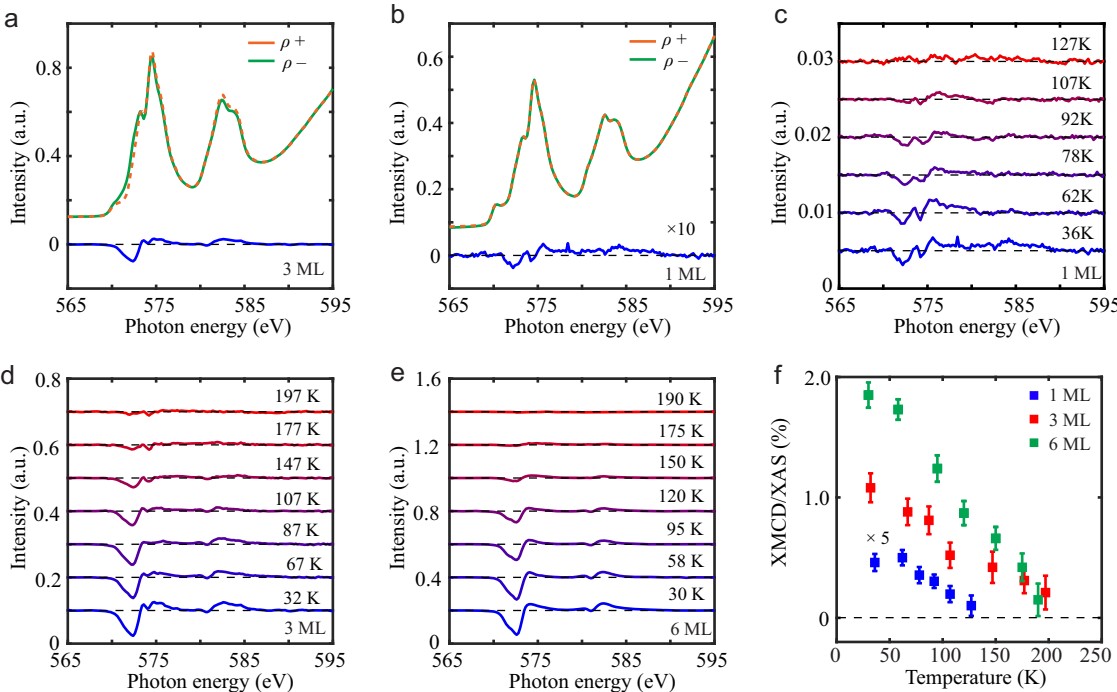

**Fig. 2 | Thickness-dependent ferromagnetism in $Cr_2Te_3$ thin films. a, b** Cr $L_{2,3}$-edge absorption spectra ($\rho^+$ and $\rho^-$) on 1 ML and 3 ML samples. The dashed orange and solid green curves denote the left-hand and right-hand polarized XAS signals, respectively. The blue curve represents the XMCD signal, which is enlarged by 10 times in 1 ML case due to the small magnitude. **c–e** Temperature-dependent XMCD spectra in 1 ML, 3 ML, and 6 ML samples. The disappearance of XMCD intensity near the $L_3$ edge is used to determine the Curie temperature $T_C$. **f** The ratio of XMCD to XAS as a function of temperature on 1 ML, 3 ML, and 6 ML samples. The error bar is the uncertainty of background estimation during the XMCD analysis.

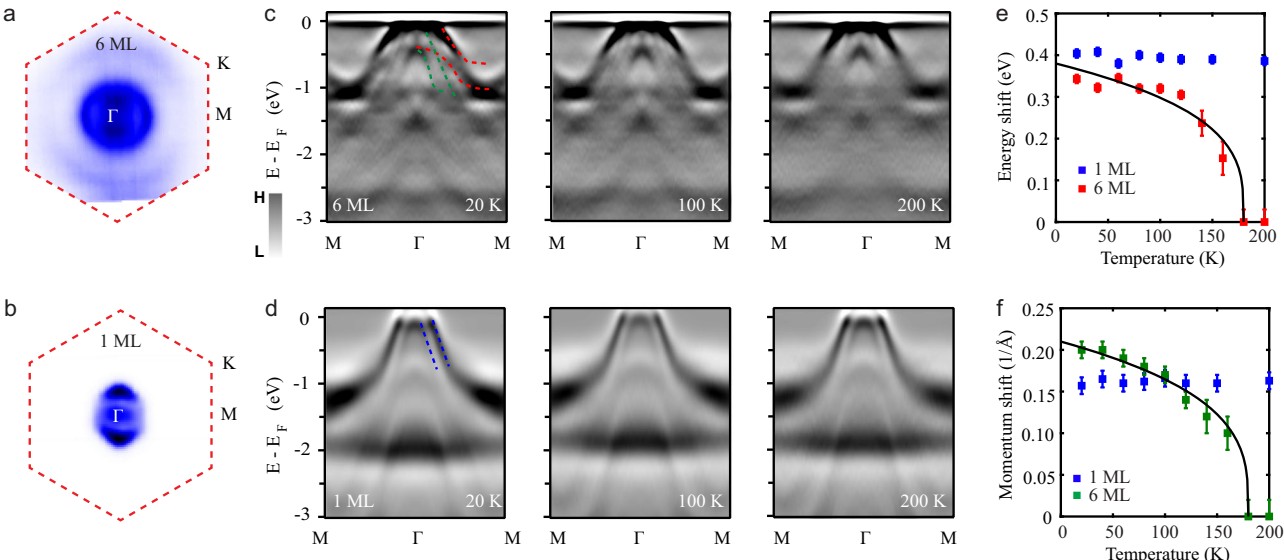

**Fig. 3 | Signature of the transition from Stoner-type to Heisenberg-type ferromagnetism in atomically thin Cr$_2$Te$_3$. a, b** Fermi surface maps of 6 ML and 1 ML samples. The integrated energy window is ±20 mV around $E_F$. **c, d** Temperature-dependent electronic structure evolution from ferromagnetic to paramagnetic state in 6 ML and 1 ML samples, respectively. Red and green dashed lines denote the Stoner exchange splitting bands in the 6 ML sample. Blue dashed lines indicate two separate bands in 1 ML sample. **e** Temperature-dependent energy shift in 6 ML and 1 ML samples. The red squares represent the energy separation of the two red dashed bands (**c**) at momentum $k = 0.4$ (1/Å). The blue squares represent the energy separation of the two blue dashed bands (**d**) at the Fermi momentum $k_F$. The black solid line is the expected energy shift from the Stoner model. The error bars are uncertainties in determining the energy distance. **f** Temperature-dependent momentum shift in 6 ML and 1 ML samples. The green squares denote the momentum distance in panel **c**. The blue squares represent the momentum separation in panel **d**. The black solid line is the expected momentum shift from the Stoner model. The error bars are uncertainties in determining the momentum distance.

investigated in 6 ML sample, as shown in Fig. 3c. Second-derivative analysis is used to enhance the visibility of low-intensity features in the band dispersion. There are explicit signatures of split Te bands below $T_C$ (green and red dashed lines). Above $T_C$, each split band merges together into one branch. Figure 3e, f summarize the temperature-dependent energy and momentum separation of the split bands. The evolution matches well with the magnetization curve predicted from the Stoner model. The detailed analysis can be found in Supplementary Fig. 4. These observations demonstrate the validity of the Stoner model for the thicker Cr$_2$Te$_3$ films.

On the contrary, the band structure of the 1 ML sample has little change across $T_C$ (Fig. 3d–f), implying the local magnetic moments persisting in the paramagnetic state. Furthermore, the local vs. itinerant features in the 1 ML and 6 ML samples are also illustrated by the spectral weight analysis for the energy distribution curve (EDC) at the $\Gamma$ point (Supplementary Fig. 5 and Fig. 6). 6 ML sample shows clear spectral weight transfer from the deeper Cr $t_{2g}$ band to the higher Te 5$p$ band, presenting strong evidence for a Stoner-like ferromagnetic exchange interaction in thicker Cr$_2$Te$_3$ films[20]. However, there is negligible spectral weight transfer in the monolayer case, implying that the itinerant scenario is insufficient to account for the 2D ferromagnetism in Cr$_2$Te$_3$.

**Electronic structure evolution from 3D to 2D**

To understand the dramatic change of $T_C$ in atomically thin Cr$_2$Te$_3$, thickness-dependent ARPES measurements were performed to illustrate the underlying electronic structures. Figure 4a displays high-resolution $s$-polarization and $p$-polarization $E$-$k$ dispersions for the 6 ML sample. The corresponding second-derivative images are shown in Fig. 4b. There are three bands crossing $E_F$ in the 6 ML sample: two dispersive hole-like bands near $\Gamma$ point come from Te 5$p$ orbitals, and one shallow band near M point stems from Cr 3$d$ orbital. The quasi-flat dispersions in the binding energy range of −1 eV to −1.6 eV are Cr $t_{2g}$ bands. Similar band structures are observed in 3 ML Cr$_2$Te$_3$ (see

Supplementary Fig. 3), explaining the same physical properties in these thicker films. First principle calculations (DFT + $U$) were carried out to understand the band structure better. By carefully tuning the correlation parameter $U$, a good match to the experimental result is achieved with $U$ in the range of 1.7−2.2 eV (Fig. 4c and Supplementary Fig. 7), which is consistent with the previous study[29].

However, the band structure changes fundamentally in the monolayer case. Only two bands pass through the $E_F$, as shown in Fig. 4d, e. The inner Te band moves upward with the band maximum touching the $E_F$. Meanwhile, the Cr $t_{2g}$ bands move downwards in the range of −1.1 eV to −1.9 eV. A simple charge transfer scenario from the substrate can be excluded due to the opposite directions of the Te and Cr bands' shifts. By considering the dimensionality and strain effect in the monolayer case, all the essential features of the band structure can be interpreted by the DFT + $U$ calculation with $U$ in the range of 1.7-3.1 eV (Fig. 4f and Supplementary Fig. 7). The similar Hubbard $U$ indicates that there is no drastic change of correlation strength in the 2D limit of Cr$_2$Te$_3$, which is in sharp contrast to the case of SrRuO$_3$[30].

One can integrate the intensity of $E$-$k$ dispersion from $\Gamma$ to M to obtain the angle-integrated photoemission spectrum (PES), which is a rough estimate of the density of states (DOS) in a material (Supplementary Fig. 8). The PES spectra of the 3 ML and 6 ML Cr$_2$Te$_3$ are nearly the same, echoing the same $T_C$ for the two samples (Fig. 4g). On the contrary, there is a significant DOS suppression near $E_F$ in the monolayer sample, implying a direct link to the suppressed $T_C$. DFT + $U$ calculations support the experimental observations: the bulk calculation is similar to the PES spectra of the 3 ML and 6 ML samples, and the monolayer curve repeats the essential features of the monolayer sample. The understanding of the electronic structure evolution from 3D to 2D is further augmented by the Cr $L_{2,3}$-edge XAS measurements (Supplementary Fig. 9). The pre-peak around 572 eV comes from the Te $M_5$ edge (Fig. 2b), reflecting the strength of covalent bonding between the Cr 3$d$ and Te 2$p$ orbitals. For example, previous studies have shown that this Te $M_5$ feature in ferromagnetic insulator

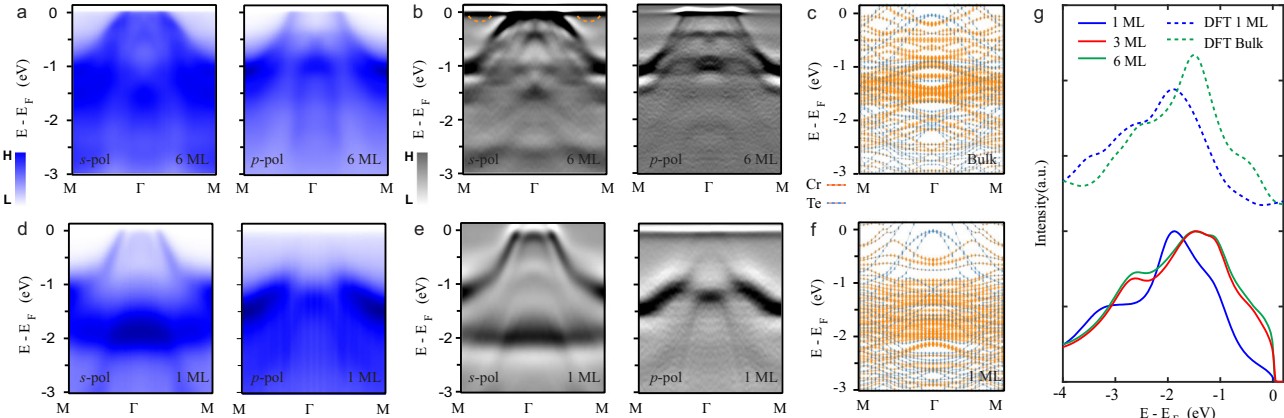

**Fig. 4 | Experimental and theoretical electronic structures in Cr$_2$Te$_3$ thin films.**
**a** $s$-polarized and $p$-polarized $E$-$k$ dispersions along $\Gamma$-M direction for 6 ML samples. The photon energies in $s$- and $p$-polarization geometries are 53 eV and 63 eV, respectively. **b** The corresponding second-derivative $E$-$k$ dispersions. The dashed orange line is the guide for the shadow band from Cr-$3d$ orbital. **c** Calculated band structures of bulk Cr$_2$Te$_3$ by the DFT + $U$ method. **d** $s$-polarized and $p$-polarized $E$-$k$ dispersions along $\Gamma$-M direction for 1 ML sample. The photon energies in $s$- and $p$-

polarization geometries are 53 eV and 63 eV, respectively. **e** The corresponding second-derivative $E$-$k$ dispersions. **f** Calculated band structures of monolayer Cr$_2$Te$_3$ by the DFT + $U$ method. **g** Comparison between the calculated density of states and the experimental photoemission spectra (integration along $\Gamma$-M direction). The DFT + $U$ method reproduces the essential features of the photoemission spectra in the 1 ML and 6 ML samples.

Cr$_2$Ge$_2$Te$_6$ is much sharper than that of ferromagnetic metal Cr$_2$Te$_3$[17,31], which is reminiscent of the thickness-dependent results from the localized regime (1 ML) to the itinerant regime (6 ML).

## Discussion

Although our study shows an obvious transition of the ferromagnetic state in the 2D limit, a Stoner model is still a good starting point to qualitatively understand the thickness-dependent ferromagnetism in atomically thin Cr$_2$Te$_3$. First, the Stoner criterion $ID(E_F) > 1$ claims $D(E_F)$ as a critical parameter to determine the stiffness of magnetic order in the ground state, where $D(E_F)$ is the DOS at $E_F$ and $I$ is the exchange parameter. The parameters $D(E_F)$ and $I$ can be obtained from the DFT + $U$ calculations of the spin-split Te $2p$ bands (Supplementary Fig. 10). The product $ID(E_F)$ is 1.59 in bulk case, confirming the validity of Stoner model in thicker Cr$_2$Te$_3$ film (Supplementary Table 1). Second, $T_C$ is dependent on $D(E_F)$ via a relation $T_C \propto \sqrt{1 - 1/ID(E_F)}$ in the Stoner model, which supports the suppressed $T_C$ in the monolayer Cr$_2$Te$_3$. Third, as the Stoner model says that the magnetization $M$ is proportional to $D(E_F)$, the much smaller XMCD percentage signal in the monolayer sample can be naturally explained.

However, the absence of exchange splitting energy in the monolayer Cr$_2$Te$_3$ is beyond the framework of the Stoner model. Actually, the product $ID(E_F)$ is 0.72 in monolayer case (Supplementary Table 1), conflicting the Stoner criterion. The strain and dimensionality are two possibilities to enact this change. The monolayer sample accompanies an octahedral to trigonal structural transformation, which splits the $t_{2g}$ triplets into $a_{1g}$ and $e_{g\pi}$ bands, with the possibility of a low spin state. But, the much smaller trigonal splitting field $Dt$ (0.2 eV) than Hubbard $U$ (1.7 eV) prohibits this scenario[32]. According to Mott's seminal paper[33], carrier density (hence bandwidth) has an anti-correlation to the lattice constant. In the monolayer case, the smaller in-plane lattice parameter should increase the carrier density, which is entirely contrary to the experimental observation. Therefore, the consistency of the results with DFT + $U$ calculations leaves the dimensionality as the only candidate for the suppressed $T_C$ and the localized ferromagnetism in the monolayer Cr$_2$Te$_3$. The bad metal property of monolayer Cr$_2$Te$_3$ may result from the interplay between the itinerant Te $5p$ bands and the localized Cr $3d$ bands. Although the energy bands near $E_F$ are mainly of delocalized Te orbitals, they can mediate the local spin moments of the Cr bands and realize long-range ferromagnetism. Intrinsically, this coherent picture belongs to the RKKY interaction,

which couples the localized and itinerant electrons in a more localized ferromagnetic metal[34]. Although previous studies report a similar $T_C$ reduction in monolayer Fe$_3$GeTe$_2$ and monolayer CrI$_3$, it has been proven to be difficult to illustrate the microscopic origin of such reduction from the electronic structures of the magnetic ground state in the true 2D limit. Our multi-probe study demonstrates an unambiguous magnetic transition from Stoner-type to Heisenberg-type in Cr$_2$Te$_3$.

Our finding provides a hint on how to increase $T_C$ in 2D ferromagnets. For Cr-Te compounds, we find that the ferromagnetic metals have a higher $T_C$ than the ferromagnetic insulators: $T_C = 170$ K in Cr$_2$Te$_3$[23], $T_C = 60$ K in Cr$_2$Ge$_2$Te$_6$[12], and $T_C = 30$ K in Cr$_2$Si$_2$Te$_6$[35]. Moreover, itinerant ferromagnets, such as Fe, Co, and Ni, possess the highest $T_C$ above 1000 K. This indicates that delocalized electrons are important to realize a higher $T_C$, which mediate the spin moments between local atoms to realize long-range order. Then, according to the Stoner model, larger $D(E_F)$ is beneficial to a higher $T_C$. $3d$ electrons of transition metal have large effective mass and narrow bandwidth, more likely to induce high $T_C$ ferromagnets if the chemical potential is tuned to approach the $3d$ flat band. Such properties have been used to control the emergent ferromagnetism in twisted bilayer graphene[36].

In conclusion, we have demonstrated the dual nature of electrons in an atomically thin ferromagnet Cr$_2$Te$_3$. Even in the strict 2D limit, the monolayer Cr$_2$Te$_3$ still retains the ferromagnetism despite the reduction in $T_C$ due to the dimensionality effect. Our multi-probe spectroscopic analysis reveals a ferromagnetic ground state transition from the itinerant to the localized picture. These results provide crucial information to optimize the performance of 2D ferromagnetism for future applications.

## Methods

### Thin film growth and ARPES

Atomically thin Cr$_2$Te$_3$ samples were grown in a custom-built MBE chamber with a base pressure of $2 \times 10^{-10}$ Torr. The samples were transferred directly to the analysis chamber for in-situ ARPES measurements at the HERS endstation of Beamline 10.0.1, Advanced Light Source, Lawrence Berkeley National Laboratory. Both $n$- and $p$-doped Si (111) substrates were used for the epitaxial growth. The $7 \times 7$ reconstructed surface of Si (111) was obtained by 10 cycles of flash annealing at 1200 °C. High-purity Cr and Te sources were simultaneously evaporated onto the Si (111) substrate that was kept at 300 °C

during the growth. The flux ratio Cr/Te = 1:4 was determined by the quartz crystal microbalance (QCM) measurement before the growth. The growth process was monitored by in situ reflective high energy electron diffraction (RHEED), and the corresponding rate is about 20 min per monolayer. After growth, the sample was annealed at 320 °C for 10 min for improved sample quality. The ARPES data were collected with a Scienta R4000 analyzer. The photon energy was 53 eV during measurements, with energy and angular resolution of 20 meV and 0.1°, respectively. The $p$-polarization direction was set to be 72° out of the plane of incident photons.

## STM measurements

A low-temperature STM system was used to characterize the as-grown films (Unisoku-1300). All the STM images were taken in the ultrahigh vacuum chamber ($2 \times 10^{-10}$ Torr) under liquid nitrogen temperature (77 K). Electrochemically etched tungsten tips were used after flashing and silver decoration in situ.

## Electronic structure calculations

The density-functional theory DFT + $U$[37] calculations of the structures, band structures, and density of states are performed by the VASP code[38,39]. We employ PAW pseudopotentials with GGA[40,41]. The total valence electrons treated explicitly in the DFT calculations are Cr $3p^6 d^5 4s^1$ and Te $5s^2 p^4$ electrons, described by a plane-wave basis set with cut-off energy of 550 eV. The bulk crystal structure was relaxed with an unrestricted variable cell relaxation as implemented in VASP. However, the supercell lattice constants for 1 ML structure were fixed upon the consideration of Si substrate, only the internal atomic positions are relaxed. The unit cell of the 1 ML structure has a 6% ab plane shrinking and a 13% stretch along the c axis, which are chosen according to the STM measured results. The relaxations were done with the Hubbard $U$ turned off and no spin polarization. To obtain the magnetic properties, we perform the collinear magnetic calculation and specify the initial magnetic moment of each atom as the value referred to in the article[42]. We find a ferromagnetic solution for the bulk crystal leads to the lowest energy. The initial magnetic moment was set up as ferromagnetic for 1 ML self-consistent run. All band structures were calculated using the corresponding relaxed structures. To acquire an optimal $U$, we vary $U$ from 0 to 3.1 eV for Cr in our tests. The Fermi surface and the renormalization ratio of the band structures have been adjusted to best agree with the ARPES data. The total DOS was determined by integrating the projected density of states (PDOS) corresponding to the bands below the Fermi surface.

## XAS and XMCD measurements

XMCD and XAS spectra show white line resonances at the Cr $L_{2,3}$-edges. Both experiments were performed at normal geometry with an incident angle of 90°. The samples were cooled by open-flow liquid helium. The spectra ($\rho^+$ and $\rho^-$) represent the parallel and anti-parallel alignment of the magnetization direction with the circularly polarized photon helicity vector, respectively. $\rho^+$ and $\rho^-$, which result from Cr $2p \to 3d$ dipole transitions, are divided roughly into the $L_3$ ($2p_{3/2}$) and $L_2$ ($2p_{1/2}$) regions. All spectra were acquired by the total electron yield (TEY) mode. The XMCD dichroism ($\Delta\rho = \rho^+ - \rho^-$) is the difference between the two spectra, and XAS is their summation ($\rho^+ + \rho^-$). We obtained the $\Delta\rho$ spectra by reversing the polarity (right- or left-circular) of the incident photon beam on the fixed external magnetic ($H \sim 0.5$ T) field along the surface normal direction. The degree of circular polarization was ∼ 95%. All spectroscopic experiments (XAS and XMCD) were carried out at beamline 13-3 at the Stanford Synchrotron Radiation Lightsource (SSRL).

## Data availability

Data are available in the Stanford Digital Repository at https://purl.stanford.edu/nf616ft8456.

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

## Acknowledgements

We would like to thank Emily Been, John J Rehr, and Joshua J Kas for insightful discussions. The work at Advanced Light Source, Lawrence Berkeley National Laboratory was supported by the U.S. Department of Energy (DOE), Office of Basic Energy Sciences under contract No. DE-AC02-05CH11231. Y.Z. was supported in part by an ALS Collaborative Postdoctoral Fellowship. The XAS and XMCD measurements were carried out at the SSRL, SLAC National Accelerator Laboratory, which is supported by the U.S. Department of Energy, Office of Basic Energy Sciences under contract no. DE-AC02- 76SF00515. The work at the SIMES/SLAC/Stanford is supported by the U.S. Department of Energy, Office of Basic Energy Sciences, Division of Materials Sciences and Engineering, under Contract No. DE-AC02-76SF00515. Parts of the theoretical calculations for this project were performed on the National Energy Research Scientific Computing Center (NERSC), a US Department of Energy Office of Science User Facility operated under Contract No. DE-AC02-05CH11231. C.P. acknowledges the support of the U.S. Department of Energy, Office of Science, Basic Energy Sciences under Contract No. DE-AC02-76SF00515 and Grant No. DE-SC0022216. The experiments taken at SJTU are supported by the Ministry of Science and Technology of China Grant No. 2019YFA0308600 and the Science and Technology Commission of Shanghai Municipality Grant No. 2019SHZDZX01.

## Author contributions

Y.Z., S.-K.M., and Z.-X.S. designed the research. Y.Z. led the film growth, with the aid of D.G., and performed ARPES measurements with J.H. and S.-K.M.; C.P., C.J., and T.P.D. provided theoretical support including the first-principle calculations and XAS analysis. H.H. and D.G. did the STM measurements. J.-S.L. and D.L. performed the XAS and XMCD measurements. All the authors participated in the scientific discussions. Y. Z. analyzed the data and wrote the paper with inputs from all the authors.

## Competing interests

The authors declare no competing interests.
