## [Peer Review File · Nature Communications]

Reviewers' Comments:

Reviewer #1:

Remarks to the Author:

The paper presents the investigation of the ferromagnetic properties of thin film Cr₂Te₃, utilizing comprehensive techniques, including ARPES, STM, XAS, and DFT. The authors report a significant reduction in the transition temperature in the monolayer, in correspondence with a reduced density of states. Furthermore, they claim the occurrence of a transition from Stoner to Heisenberg type. The manuscript is well-written and well-researched. However, there are a few questions that require further clarification, and the unique contributions of the present study are not entirely apparent.

Firstly, while the authors argue that the simple Stoner model cannot account for the absence of exchange splitting energy, and thus a Heisenberg type is possible, it is also worth considering the possibility that the splitting is very small and may not be discernible. Given that the band structures from ARPES experiments are not very clear, it is possible that the splitting is too small and can't be seen from the current picture resolution. It would be beneficial if the authors could provide additional evidence to support their claim of the Heisenberg type. For example, the author could consider performing calculations using both Stoner and Heisenberg models.

Secondly, the study mentions that similar phenomena of reducing transition temperatures have been reported in previous experiments [13-16]. To ensure the novelty of the present study, it would be beneficial if the authors could elaborate on what sets their findings apart from previous research.

Lastly, the authors state that the interaction strength is the same ($U=1.7$ eV) for both the 6ML and 1ML samples. However, the Extended Data Fig.6 suggests that a range of interaction strengths could explain the results. To strengthen their argument, the authors could provide an upper and lower bound of the interaction strength beyond which the model fails to accurately represent the experimental results.

Reviewer #2:

Remarks to the Author:

This paper reports the synthesis of Cr₂Te₃ thin films with different thickness using MBE methods. Angle-resolved photoemission spectroscopy, scanning tunneling microscopy, x-ray absorption spectroscopy, and first-principle calculations are performed to characterize the structure and physical properties of the thickness-dependent films. Both the 1ML and 6ML Cr₂Te₃ samples show ferromagnetism. However, the Curie temperature of the 1ML sample is reduced considerably compared with that of the 6ML one. Also, different electronic properties from localized to itinerant behaviors are found to occur in different-layer samples. A ferromagnetic transition from Stoner to Heisenberg-type is claimed to be observed in the atomically thin limit, suggesting that dimensionality is an effective tuning knob to manipulate the properties of 2D magnetism. The results are potentially interesting in condensed matter physics and magnetic material sciences. However, I cannot recommend for possible publication for the current version. Major revisions are necessary for possible publication in Nat. Commun. My detailed comments are shown as follows.

1. In the introduction, the authors described different theoretical models and mentioned some 2D magnetic materials. However, there almost is no introduction for previous studies on 2D Cr₂Te₃ materials. Also, the authors did not compare the current data with those of other reported results of 2D Cr₂Te₃ thin films prepared using different methods with different thickness. I thus strongly suggest to add some background descriptions for the magnetism of Cr₂Te₃ thin films and make related comparison in the main text.

2. In Fig. 1c, there is a clear diffraction peak around 22 deg. What's the origin of this peak? Unknown impurity? How about the stability of the 6ML and 1ML samples in atmosphere?

3. As shown in Fig 2a and b, the spectral shape of XAS for 3ML and 1ML samples is different. For example, the pre-edge peak around 570 eV is sharp for 1ML but broad for 3ML. Also, the relative intensity for the two peaks in L2 edge is clearly different for these two samples. What conclusion

can be made from the distinct XAS shapes? As is well known, a chemically stoichiometric Cr_2Te_3 thin film is difficult to obtain. Do the different XAS shapes mean somewhat different chemical compositions?

4. For FM materials, one can find significant XMCD signals. However, based on the weak XMCD signal, it is not safe to conclude that the material is ferromagnetism. As is well known, a magnetic field is used to get the XMCD data. In this case, field induced FM signals are possible to occur even for an antiferromagnetic material with slightly canted spins. To further confirm the intrinsic FM features for the 1 and 6 ML samples, temperature dependent magnetic susceptibility and field dependent isothermal magnetization measures are needed.

5. As shown in Fig 1f, different CrTe_6 polyhedral units are proposed. Can the authors provide the approximate values for Cr-Te bond lengths in these two units?

6. To further confirm the different electronic properties from localized to itinerant, is possible to perform electrical transport measurements, such as temperature dependent resistivity, for the 1 and 6 ML samples?

7. Some explanations are too simple to be understood for readers. For example (but not limited), "Temperature-dependent electronic structure evolution from the ferromagnetic state to the paramagnetic state on the 6 ML sample is shown in Fig. 3c". I suggest to give more detailed descriptions to explain the result.

Reply to Reviewer's Comments

Manuscript NCOMMS-22-49746-T: "From Stoner to Local Moment Magnetism in Atomically Thin Cr₂Te₃" by Yong Zhong et al.

We would like to thank the reviewers for their comments and suggestions, which have helped us clarify some important aspects of our results and improve the presentation of the manuscript. Please find below our response to all the Reviewers' comments and the corresponding revisions that we have made to the manuscript. The original comments from the Reviewers are labeled in blue, our response is in black, and our action is summarized in red. The changes that we made are also summarized in the list of changes at the end of this document.

Reviewer 1

Comments:

The paper presents the investigation of the ferromagnetic properties of thin film Cr₂Te₃, utilizing comprehensive techniques, including ARPES, STM, XAS, and DFT. The authors report a significant reduction in the transition temperature in the monolayer, in correspondence with a reduced density of states. Furthermore, they claim the occurrence of a transition from Stoner to Heisenberg type. The manuscript is well-written and well-researched. However, there are a few questions that require further clarification, and the unique contributions of the present study are not entirely apparent.

Reply: We thank Reviewer #1 for the thoughtful review of our manuscript and for the positive assessment of our work. Reviewer #1 made a couple of comments to improve our manuscript. Below, we consider these suggestions and highlight the corresponding changes we have made to the manuscript.

1. Firstly, while the authors argue that the simple Stoner model cannot account for the absence of exchange splitting energy, and thus a Heisenberg type is possible, it is also worth considering the possibility that the splitting is very small and may not be discernible. Given that the band structures from ARPES experiments are not very clear, it is possible that the splitting is too small and can't be seen from the current picture resolution. it would be beneficial if the authors could

provide additional evidence to support their claim of the Heisenberg type. For example, the author could consider performing calculations using both Stoner and Heisenberg models.

Reply: We agree with Reviewer #1 that theoretical calculations can provide more detailed evidence to support the Stoner-type to Heisenberg-type transition as the film thickness approaches the monolayer limit. We performed J_{ij} related calculations for the revised manuscript, and the details are summarized below.

(i) In our original manuscript, we checked the validity of the Stoner model for Cr_2Te_3 thin films by analyzing the exchange splitting energy of Te 2p bands (Fig. 3). Here, the spin-resolved Te bands are calculated by the DFT + U methods (Fig. R1), from which we can extract the density of states at Fermi level $D(E_F)$ and Stoner parameter I . Table R1 shows the details of the parameter we obtain from the calculation. For the bulk case, the calculated exchange splitting energy is 240 meV, which is comparable to the value from our ARPES results (330 meV). The product $ID(E_F)$ is larger than 1, satisfying the Stoner criterion. In contrast, the calculated exchange splitting in the monolayer case is 180 meV, which should be observed in the ARPES measurements since the energy resolution of our measurement is around 20 meV. The absence of splitting in our measured Te bands, therefore, implies the invalidity of the Stoner model in monolayer Cr_2Te_3 , which is further confirmed by the small value of $ID(E_F)$.

(ii) Although the DFT + U calculations in our paper are based on the Stoner model, we can still map it into the Heisenberg model with the following Hamiltonian:

$$H = \sum_{i,j} J_{ij} S_i S_j + \sum_i A (S_i^z)^2$$

where S_i is the spin operator on site i , and J_{ij} is the exchange coupling between spins on site i and j ; A is the perpendicular anisotropy parameter, which is essential to sustain the ferromagnetic order in the 2D limit. A negative J_{ij} corresponds to the ferromagnetic interaction. Larger magnitude of J_{ij} indicates more localized electrons in the system. The calculations of J_{ij} is beyond our capability due to the complex band structure of Cr_2Te_3 . But we can obtain valuable information by carefully inspecting the J_{ij} parameters in Fe_3GeTe_2 . This comparison is reasonable since the resistivity measurement shows a similar itinerant-to-localized ferromagnetic transition in monolayer Fe_3GeTe_2 [*Nature* **563**, 94 (2018)]. The parameter J_{ij} of monolayer film is much larger than that of bulk samples, implying the more localized property of electrons in 2D Fe_3GeTe_2 .

(iii) We added sentences “The parameters $D(E_F)$ and I can be obtained from the DFT + U calculations of the spin-split Te 2p bands (Supplementary Fig. 10). The product $ID(E_F)$ is 1.59 in bulk case, confirming the validity of Stoner model in thicker Cr_2Te_3 film (Supplementary Table 1).” and “Actually, the product $ID(E_F)$ is 0.72 in monolayer case (Supplementary Table 1), conflicting the Stoner criterion.” in the main text to address the above issues. Table R1 and Fig.R1 are added in the new manuscript as supplementary information to make it more explicit.

Fig. R1 Calculated density of states of Te bands for 1 ML (a) and bulk Cr_2Te_3 (b).

Table R1 Parameters in the Stoner model

Thickness	$D(E_F)$	$ E(p \uparrow) - E(p \downarrow) $	I	$ID(E_F)$
1ML	1.61	0.18	0.45	0.72
Bulk	2.78	0.24	0.57	1.59

2. Secondly, the study mentions that similar phenomena of reducing transition temperatures have been reported in previous experiments [13-16]. To ensure the novelty of the present study, it would be beneficial if the authors could elaborate on what sets their findings apart from previous research.

Reply: We thank Reviewer #1 for the valuable suggestion. It is true that the previous experiments, such as Ref. 13 – 16, reported similar reductions in the transition temperature, but none of them has provided a detailed investigation on the microscopic origin of such reduction from the electronic structure point of view. To stress the novelty of our study more explicitly, we have added the following sentences in our revised manuscript. “Although previous studies report a

similar T_C reduction in monolayer Fe_3GeTe_2 and monolayer CrI_3 , it has been proven to be difficult to illustrate the microscopic origin of such reduction from the electronic structures of the magnetic ground state in the true 2D limit. Our multi-probe study demonstrates an unambiguous magnetic transition from Stoner-type to Heisenberg-type in Cr_2Te_3 .”

3. Lastly, the authors state that the interaction strength is the same ($U=1.7$ eV) for both the 6ML and 1ML samples. However, the Extended Data Fig.6 suggests that a range of interaction strengths could explain the results. To strengthen their argument, the authors could provide an upper and lower bound of the interaction strength beyond which the model fails to accurately represent the experimental results.

Reply: We thank Reviewer #1 for pointing it out. In the new version of the manuscript, we now provide the upper and lower bounds of Hubbard U : $U = 1.7-2.2\text{eV}$ for bulk and $U=1.7-3.1\text{eV}$ for monolayer. We have also modified the related statements in the main text and highlighted them in red.

Reviewer 2

Comments:

This paper reports the synthesis of Cr_2Te_3 thin films with different thickness using MBE methods. Angle-resolved photoemission spectroscopy, scanning tunneling microscopy, x-ray absorption spectroscopy, and first-principle calculations are performed to characterize the structure and physical properties of the thickness-dependent films. Both the 1ML and 6ML Cr_2Te_3 samples show ferromagnetism. However, the Curie temperature of the 1ML sample is reduced considerably compared with that of the 6ML one. Also, different electronic properties from localized to itinerant behaviors are found to occur in different-layer samples. A ferromagnetic transition from Stoner to Heisenberg-type is claimed to be observed in the atomically thin limit, suggesting that dimensionality is an effective tuning knob to manipulate the properties of 2D magnetism. The results are potentially interested in condensed matter physics and magnetic material sciences. However, I cannot recommend for possible publication for the current version. Major revisions are necessary for possible publication in Nat. Commun. My detailed comments are shown as follows.

Reply: We appreciate Reviewer #2's recognition that our results are "potentially interested" in condensed matter physics and magnetic material sciences. At the same time, we do agree with the Reviewer's concern that major revisions are necessary to clarify some important issues, such as the determination of the structural and magnetic phase of Cr_2Te_3 , the interpretation of XAS spectra, and the transport and magnetization measurements. Below are our point-by-point responses.

1. In the introduction, the authors described different theoretical models and mentioned some 2D magnetic materials. However, there almost is no introduction for previous studies on 2D Cr_2Te_3 materials. Also, the authors did not compare the current data with those of other reported results of 2D Cr_2Te_3 thin films prepared using different methods with different thickness. I thus strongly suggest to add some background descriptions for the magnetism of Cr_2Te_3 thin films and make related comparison in the main text.

Reply: We thank Reviewer #2's suggestions. We have added some background descriptions for the previous studies of Cr_2Te_3 thin films in the Introduction part and compared our results with them in the main text. Below are the details.

(i) Magnetic Cr_2Te_3 thin films have been prepared by the MBE method on Si (111) and Al_2O_3 (0001) substrates [*ACS Nano* **9**, 3772 (2015); *Nat. Commun.* **14**, 3222 (2023)]. The Curie temperature ($T_C = 170$ K) is the same as that of Cr_2Te_3 single crystal, as the film thickness exceeds 4 nm. Correspondingly, the sentence " Cr_2Te_3 thin films have been fabricated by molecular beam epitaxy (MBE) method on silicon and sapphire substrates [25,26], displaying a comparable T_C to the single crystals as the film thickness exceeds 4 nm" is added in the Introduction.

(ii) We also used MBE to fabricate atomically thin Cr_2Te_3 films in this study. By performing superconducting quantum interference device (SQUID) measurement, the Curie temperature of the 6 ML sample is determined as $T_C = 170$ K, the same as that in previous literatures [*ACS Nano* **9**, 3772 (2015); *Nat. Commun.* **14**, 3222 (2023)]. Therefore, the sentence "Temperature-dependent magnetization of 6 ML sample was measured by superconducting quantum interference device (SQUID) in Supplementary Fig. 2, showing the same T_C as that in literatures [25,26]" is added in the main text.

(iii) Having benchmarked the magnetic properties of Cr_2Te_3 in a 6ML sample, we further explore the 2D ferromagnetism in the monolayer limit, which sets our study apart from the previous research. Correspondingly, the sentence "However, to our best knowledge, there is no

exploration of the ferromagnetism in the atomically thin 2D limit ” is added in the Introduction to address the novelty of our research.

2. In Fig. 1c, there is a clear diffraction peak around 22 deg. What’s the origin of this peak? Unknown impurity? How about the stability of the 6ML and 1ML samples in atmosphere?

Reply: We thank Reviewer #2 for the comment. There is no diffraction peak around 22 deg in the original XRD curve, as shown in Fig. R2a. Instead, there is a feature near 27 deg. We assume that Reviewer #2 questioned the origin of this peak. In order to obtain a clear signal of characteristic peaks, we used the samples with 20 nm thickness in the original version of the manuscript (Fig.R2a), nearly three times that of the 6 ML sample (7 nm). The diffraction peak around 27 deg comes from the (100) plane of Cr_2Te_3 , which has been clearly illustrated in the previous study of $\text{Sb}_2\text{Te}_3/\text{Cr}_2\text{Te}_3$ heterostructure [*Sci. Rep.* **9**, 10793 (2019)].

This is due to the reduced strain effect from the substrate as the film becomes thicker. However, in our study, we are only interested in the atomically thin Cr_2Te_3 with a thickness less than 7 nm. It is better to directly measure the XRD of the 6 ML sample. We have made additional XRD measurements directly on 6 ML samples. As shown in Figs. R2b and c, our results demonstrate the pure phase of Cr_2Te_3 along the c-axis direction, confirming that the epitaxial growth mode persists at least up to 7 nm. In our revised version, we have replaced Figure 1c with the new result from the 6 ML sample. The Cr_2Te_3 thin films are very sensitive to the atmosphere. We need to deposit amorphous tellurium as the capping layer to protect the samples.

Fig. R2 (a) The original XRD curve in the manuscript. The film thickness is 20 nm. The peak around 27 deg is the (100) plane of Cr_2Te_3 . (b, c) XRD measurements on 6 ML samples using rocking curve and triple axis modes.

3. As shown in Fig 2a and b, the spectral shape of XAS for 3ML and 1ML samples is different. For example, the pre-edge peak around 570 eV is sharp for 1ML but broad for 3ML. Also, the relative intensity for the two peaks in L_2 edge is clearly different for these two samples. What conclusion can be made from the distinct XAS shapes? As is well known, a chemically stoichiometric Cr_2Te_3 thin film is difficult to obtain. Do the different XAS shapes mean somewhat different chemical compositions?

Reply: We agree with Reviewer #2 that the spectral shape of XAS for 3ML and 1ML samples is different. In fact, we can obtain useful electronic information by analyzing the details of XAS spectra. Below are the details.

(i) The pre-peak around 572 eV comes from the Te M_5 peak, which is absent in the atomic multiplet theory calculations of the Cr $L_{2,3}$ XAS curve (shown in the lower panel of Fig. R3a). This is reasonable because the atomic multiplet theory only deals with the local structure of the Cr 3d bands and does not consider the contribution from the Te orbital. In fact, the intensity of the Te M_5 peak is anticorrelated with the covalent mixing between the Cr 3d and Te 2p orbitals. For example, the Te M_5 peak of the ferromagnetic insulator $\text{Cr}_2\text{Ge}_2\text{Te}_6$ is quite sharp [*Phys. Rev. B* **101**, 205125 (2020)], while this feature is much weaker in the ferromagnetic metal Cr_2Te_3 (Fig. R3) [*Sci. Rep.* **9**, 10793 (2019)]. In our study, there is a clear thickness-dependent evolution of the Te M_5 peak from the localized regime (1 ML sample) to the itinerant regime (6 ML sample), as shown in Supplementary Fig. 9. Conclusively, the magnitude of the Te M_5 peak is closely related to the localized property of electrons in Cr_2Te_3 thin films.

(ii) The relative intensity of the two peaks at the L_2 edge further provides strong evidence of thickness-dependent metallicity in our samples. As shown in Fig. R3, the relative ratio in Cr_2Te_3 is much larger than that of $\text{Cr}_2\text{Ge}_2\text{Te}_6$, quite similar to the thickness-dependent results in Supplementary Fig. 9. This observation confirms the itinerant to localized ferromagnetic transition in the monolayer limit.

(iii) The different XAS shapes don't mean other chemical compositions in our samples. Our XRD measurements demonstrate a pure Cr_2Te_3 phase.

(iv) In order to explain the important role of XAS spectral details in understanding the electronic information, we have now added the sentences in the main text. "The pre-peak around 572 eV comes from the Te M_5 edge (Fig. 2b), which reflects the strength of covalent bonding between the Cr 3d and Te 2p orbitals. For example, previous studies have shown that the Te M_5 feature in ferromagnetic insulator $\text{Cr}_2\text{Ge}_2\text{Te}_6$ is much sharper than that of ferromagnetic metal Cr_2Te_3

[17,31], which is reminiscent of the thickness-dependent results from the localized regime (1 ML) to the itinerant regime (6 ML).”

Fig. R3 (a) Upper panel: dichroic absorption spectra for the Cr $L_{2,3}$ edges in $\text{Cr}_2\text{Ge}_2\text{Te}_6$ [17]. Lower panel: simulation of XAS and XMCD from atomic multiplet theory. (b) Dichroic absorption spectra for the Cr $L_{2,3}$ edges in Cr_2Te_3 [31].

4. For FM materials, one can find significant XMCD signals. However, based on the weak XMCD signal, it is not safe to conclude that the material is ferromagnetism. As is well known, a magnetic field is used to get the XMCD data. In this case, field induced FM signals are possible to occur even for an antiferromagnetic material with slightly canted spins. To further confirm the intrinsic FM features for the 1 and 6 ML samples, temperature dependent and field dependent isothermal magnetization measures are needed.

Reply: We agree with Reviewer #2 that temperature-dependent magnetic susceptibility and field-dependent isothermal magnetization measurements are a more straightforward method to confirm intrinsic ferromagnetism. We also agree with the Review that XMCD alone may not be sufficient evidence to show the ferromagnetism in this material. To provide additional proof of ferromagnetism, we have performed the SQUID measurements on 6 ML samples to extract these physical quantities. Fig. R4a is the temperature-dependent magnetic moment, from which $T_C = 170$ K can be verified. Fig. R4b is the field-dependent isothermal magnetization. Based on these

measurements, the intrinsic ferromagnetism is confirmed for the 6 ML sample. In our revised version, we have added these figures in the Supplementary Information to provide additional evidence of ferromagnetism.

Fig. R4 (a) Temperature-dependent magnetic moment. (b) Field-dependent isothermal magnetization.

For 1 ML sample, the elusive ferromagnetic signal requires a technique of much greater sensitivity than that provided by conventional magnetometers (such as SQUID) due to the intrinsic weak signal from the monolayer. That’s the reason why previous literature usually uses magneto-optical techniques (such as XMCD and Kerr measurements) to explore the 2D ferromagnetism (see Table R2). Based on this, we added the sentence “However, the elusive ferromagnetic signal of 1 ML sample requires a technique of better sensitivity than that provided by conventional SQUID” in the main text to explain why we use the XMCD method to explore the tiny ferromagnetism in the monolayer sample.

Table R2 Methods to detect 2D ferromagnetism

2D ferromagnets	Detection method	References
CrI ₃	Kerr measurement	Nature 546 , 270 (2017)
Cr ₂ Ge ₂ Te ₆	Kerr measurement	Nature 546 , 265 (2017)
Fe ₃ GeTe ₂	XMCD, anomalous Hall resistance	Nature 563 , 94 (2018)
Fe ₃ GeTe ₂	XMCD, Kerr measurement	Nat. Mater. 17 , 778 (2018)
CrI ₃	XMCD	Science 374 , 616 (2021)
CrTe ₂	XMCD	Nat. Commun. 12 , 2492 (2021)

5. As shown in Fig 1f, different CrTe₆ polyhedral units are proposed. Can the authors provide the approximate values for Cr-Te bond lengths in these two units?

Reply: Yes, we can calculate the Cr-Te bond lengths in these two unit cells. We denote in-plane and out-of-plane lattice constants as a and c , the nearest Te-Te bond length as d_{Te-Te} , the height between Cr plane and Te plane as h_{Te-Te} , and the nearest Cr-Te bond length as d_{Cr-Te} . Following relations can be used to obtain d_{Cr-Te} :

$$d_{Te-Te} = a/3$$

$$h_{Te-Te} = c/8$$

$$d_{Cr-Te} = \sqrt{d_{Te-Te}^2 + h_{Cr-Te}^2}$$

The bond lengths d_{Cr-Te} are 2.74 Å and 2.68 Å in octahedral and trigonal structures, respectively. Table R3 shows the detailed information.

Table R3 Lattice information

Sample thickness	$a / \text{Å}$	$c / \text{Å}$	$d_{Te-Te} / \text{Å}$	$h_{Te-Te} / \text{Å}$	$d_{Cr-Te} / \text{Å}$
6ML	6.81	12.4	2.27	1.55	2.74
1ML	6.32	13.3	2.11	1.66	2.68

6. To further confirm the different electronic properties from localized to itinerant, is possible to perform electrical transport measurements, such as temperature dependent resistivity, for the 1 and 6 ML samples?

Reply: We thank the Referee for raising this point. Fig. R5a is the temperature-dependent resistivity measurement of 6 ML sample on Al₂O₃ substrate. The Curie temperature T_C is around 165 K, within the error bar compared to the previous results (upper panel of Fig. R5c).

Considering the n-type Si substrate acts as a conductive channel, we could not obtain the intrinsic resistance of monolayer Cr₂Te₃ in transport measurements (Fig. R5b shows the resistivity curve of 1ML sample on Si substrate. Most of the signal comes from the n-type Si) unless the

film thickness exceeds tens of nanometers (see the lower panel of Fig. R5c). Additionally, the monolayer Cr_2Te_3 sample cannot be epitaxially grown on Al_2O_3 substrate due to lattice mismatch, making it difficult to measure the resistivity behavior in the monolayer limit.

Although we have no direct evidence from transport measurements to demonstrate the localized electrons in monolayer film, we can gain useful information from the thickness-dependent resistivity of exfoliated Fe_3GeTe_2 flakes [*Nature* **563**, 94 (2018)], as shown in Fig. R5d. While the 6 ML sample exhibits metallic curve, the monolayer flake displays a standard insulating property with localized electrons. Since Fe_3GeTe_2 is also a ferromagnetic metal, it is reasonable to expect similar behavior in atomically thin Cr_2Te_3 films.

Fig. R5 (a) Resistivity measurement for 6 ML Cr_2Te_3 on Al_2O_3 substrate. (b) Resistivity measurement for 1 ML Cr_2Te_3 on Si substrate. (c) Upper panel: 4 nm Cr_2Te_3 on Al_2O_3 substrate. Lower panel: 20 nm Cr_2Te_3 on Si substrate [25]. (d) Thickness-dependent resistivity measurement for exfoliated Fe_3GeTe_2 flakes [15].

7. Some explanations are too simple to be understood for readers. For example (but not limited), “Temperature-dependent electronic structure evolution from the ferromagnetic state to the paramagnetic state on the 6 ML sample is shown in Fig. 3c”. I suggest to give more detailed descriptions to explain the result.

Reply: We appreciate the Reviewer’s careful reading of our manuscript. We have carefully reviewed our manuscript to expand the explanation for our readers. We have added multiple sentences in this regard, as shown in the early part of our response letter and summarized in the list of changes below. For the particular sentence that the Reviewer pointed out, we have replaced the sentence “Temperature-dependent electronic structure evolution from the ferromagnetic state to the paramagnetic state on the 6 ML sample is shown in Fig. 3c” as “By varying the temperature below and above T_C , the electronic structures of the ferromagnetic and paramagnetic states are systematically investigated in 6 ML sample, as shown in Fig. 3c” to make the statement more explicit.

List of Changes:

Changes in the main text:

1. Line 75. We have added some background descriptions of the Cr_2Te_3 thin films grown by the MBE technique, as suggested by Reviewer #2.
2. Line 77. We have added the statement regarding the lack of exploration of 2D ferromagnetism in Cr_2Te_3 to address the significance of our research.
3. Line 108. We have compared the magnetic property of our thin films with that of previous literature, following the suggestions of Reviewer #2.
4. Line 110. We have added the sentence to address the uniqueness of magneto-optical techniques to explore the 2D magnetism.
5. Line 138. We have modified the statement to make it more explicit, as recommended by Reviewer #2.
6. Line 166 and 173. We have given the upper and lower bounds of Hubbard U for the bulk and monolayer Cr_2Te_3 , as Reviewer #1 suggested.
7. Line 184. We have expanded the analysis of the pre-peak feature around 572 eV to illustrate the thickness-dependent electronic structure in Cr_2Te_3 following the suggestions of Reviewer #2.

8. Line 194 and 201. We have added the comments on the theoretical calculations of exchange splitting energy for the bulk and monolayer Cr_2Te_3 , as recommended by Reviewer #1.
9. Line 215. We have revised the manuscript to stress the novelty of our research following the suggestions of Reviewer #1.
10. Line 378. We have replaced Fig. 1c with new results from 6 ML film to confirm the pure Cr_2Te_3 phase of our samples in response to Reviewer #2's comments.
11. We have updated Refs. 26 and 31 to make them compatible with the new manuscript.

Changes in the supplementary information:

1. Added Supplementary Fig. 2 to display the magnetization measurements of 6 ML samples in response to Reviewer #2's comments.
2. Added Supplementary Fig. 10 and Supplementary Table 1 to show the calculated Stoner-model parameters in response to Reviewer #1's comments.

Reviewers' Comments:

Reviewer #1:

Remarks to the Author:

I have reviewed the authors' responses to my comments, and I am pleased to see they have addressed all of the concerns in a comprehensive and satisfactory manner.

The authors have enhanced the manuscript by providing a more in-depth investigation into the transition from the Stoner model to the Heisenberg model as the film thickness approaches the monolayer limit. Furthermore, I appreciate the effort to elucidate the novelty of the study compared to previous research.

In light of these modifications, I find the paper to be well-crafted and the research significant, shedding new light on the ferromagnetic properties of thin film Cr₂Te₃. Given the thoroughness of the revisions and the quality of the work, I recommend the paper for publication.

Reviewer #2:

Remarks to the Author:

The authors revised the manuscript based on referees' comments and suggestions. The revised manuscript is now acceptable for publication in Nature Communications.

Reply to Reviewers' Comments:

Thank you for sending us the Reviewers' reports and for offering us the opportunity to publish our manuscript. Please find below our response to all the Reviewers' comments. The original comments from the Reviewers are labeled in blue, our response is in black.

Reviewer 1

I have reviewed the authors' responses to my comments, and I am pleased to see they have addressed all of the concerns in a comprehensive and satisfactory manner. The authors have enhanced the manuscript by providing a more in-depth investigation into the transition from the Stoner model to the Heisenberg model as the film thickness approaches the monolayer limit. Furthermore, I appreciate the effort to elucidate the novelty of the study compared to previous research. In light of these modifications, I find the paper to be well-crafted and the research significant, shedding new light on the ferromagnetic properties of thin film Cr₂Te₃. Given the thoroughness of the revisions and the quality of the work, I recommend the paper for publication.

Reply:

We thank Reviewer #1 for recommending our manuscript to be published in the present form.

Reviewer 2

The authors revised the manuscript based on referees' comments and suggestions. The revised manuscript is now acceptable for publication in *Nature Communications*.

Reply:

We thank Reviewer #2 for recommending our manuscript for the publication in *Nature Communications*.